# Transparency for Participation through the Communication Approach

**Pedro Molina Rodríguez-Navas** [1,*] **, Narcisa Medranda Morales** [2] **and Johamna Muñoz Lalinde** [3]

1   Department of Audiovisual Communication and Advertising, Universidad Autónoma de Barcelona, 08193 Barcelona, Spain
2   School of Social Communication, Universidad Polítécnica Salesiana, Quito 170517, Ecuador; nmedranda@ups.edu.ec
3   Mass Communication and Journalism, Universidad del Norte, Barranquilla 080001, Colombia; lalinde@uninorte.edu.co
*   Correspondence: pedro.molina@uab.cat

**Abstract:** Transparency is a communicative process whose aim is to provide citizens with information that will promote their participation in public affairs. However, its application is often reduced to a legally stipulated administrative act. In contrast, this article sets out the principles, attributes and evidence of transparency from a communication perspective, taking into consideration that transparency is treated as a process through which recipients obtain, understand and use information. This study focuses on the transparency of local town councils, although most of the concepts could be applied at other levels of public administration. To establish this framework, the legislation and application of transparency in three countries (Spain, Ecuador and Colombia) was studied using the Infoparticipa method designed with a communication approach in mind. A comparative study was then carried out using methods designed in other disciplines. Through this approach, the benefits of transparency were categorized to define six principles—disclosure, strengthening, visibility, comprehensibility, dissemination and humanism—and eight attributes of transparent information: veracity, timeliness, accessibility, usability, intelligibility, universality, pluralism and plurality. For each attribute, the evidence of its application was determined. This framework clarifies the perspective of transparency for participation from a communication approach.

**Keywords:** transparency; participation; public communication; institutional communication; digital media; local administration

## 1. Introduction

Transparency can be approached from disciplines such as law, economics, political science and archival or public administration studies. However, in order to approach it correctly, we need to bear in mind that without public communication there is no transparency. Therefore, transparency is not an administrative act of regulatory compliance but a process in continual progress, since public institutions constantly generate new information that must be communicated while it is current, relevant and useful. Thus, treating transparency as a communicative process, the recipients of information are made visible and are given a determining role since it has been established that without their participation there is no transparency. In this way, important questions can begin to be asked: who are the target audiences? why should local councils address them? what is the best way to communicate information to recipients? and, how and for what purposed are they expected to use it?

Transparency conceived thus in the context of current digital and online communication [1] involves positioning both users as consumers of information and other active recipients, prosumers and emirecs [2], who redistribute information, re-writing it and contributing knowledge, opinions and proposals, thus overcoming the margins of local

government communication departments and the tools local governments provide for participation. María José Canel and Vilma Luoma-aho [3] presented how changes in demands, expectations, forms of communication, roles and even the very nature of citizenship affect the public sector. Thus, new challenges arise such as responding to demands or achieving citizen engagement with institutions, to achieve an increase in intangibles aspects such as reputational or a greater perception of legitimacy.

Thus, institutions have to address these communication flows in order to design and adequately and permanently reconstruct their objectives and messages, incorporating the results of this interactive participatory process [4] without forgetting that the same local government information can be used as starting material by media organizations in the creation of news stories.

Framing the issue of transparency from a communication perspective, we should avoid entering into considerations of public communication as a whole. For this, we must limit the subject of this study, namely transparency, to information about the government and the management carried out by public institutions (which are the focus of this work, excluding other types of private organizations). It should be noted that this definition includes information that is managed in order to be able to carry out the institutional purposes, exercise them and obtain results that must be collected through quality indicators on the services provided. Moreover, public institutions must demonstrate compliance with all legal precepts in order to carry out their actions, which includes doing so in accordance with public deadlines and procedures.

However, the purposes traditionally associated with transparency include the prevention of corruption, the improvement of good governance, the facilitation of accountability and the promotion of citizen participation. All are plausible objectives but to achieve them transparency is an essential element, though not enough in itself [5–7]. However, considering transparency from the perspective of communication, the primary issue of the problem is addressed, namely that the fundamental aim of transparency is to facilitate and stimulate the participation of citizens in the democratic system. Thus, the Transparency Law of Catalonia (Art. 2. a.) establishes that transparency is a proactive action to publicize information in a way that facilitates participation in public affairs [8].

This communicative concept is reinforced as ICTs have undoubtedly fostered interest in transparency and have increased the possibilities for applying it. In 1980, Michel Destot [9] highlighted the effort of the city of Grenoble to get closer to its citizens, counting on journalists-correspondents and information assistants in a project consisting of 50 public centers equipped with telephones, televisions and computers. Despite these commendable efforts, it was the advent of digital technologies and the Internet in the 21st century that enabled a qualitative leap forward to be made, providing citizens with information and permanent communication channels that were accessible from any fixed or mobile device [10–12]. However, public administrations do not always have all of these resources [13–15].

This change led to a new perspective on transparency that, in its application, saw a turning point with President Obama's Memorandum [16], which determined that the government must be transparent and participatory, driving a global transparency movement. At the same time, new concepts such as open government, open data, e-government or e-democracy began to appear [17–19], all of them related to the availability of information, online communication channels and the promotion of democratic participation. Moreover, along with the work of public institutions, other organizations use so-called civic technologies to involve citizens in political decision-making [20], with government transparency being one of the most important objectives of those organizations that develop and use them [21]. Thus, the questions we asked at the beginning of this article are posed in the context of a demand for deepening democracy and the technological possibilities to promote it.

On the other hand, Etzioni [22] stated that—in the best of cases—the supposed benefits of transparency are limited, and the usefulness of the information does not compensate

for its high cost. Fenster [23] also argued that theory about transparency has not been sufficiently developed and that it starts from assumptions that must still be clarified to get rid of idealized concepts not tested in practice. Alloa and Thöma [24] mentioned that we can talk of the field of Critical Transparency Studies, which has emerged due to the abusive and ambiguous use of the term 'transparency' and because it has also been noted that public administrations often use it as a marketing strategy rather than as an effective strategy.

In this work, it is not assumed that transparency by itself solves all kinds of problems related to the lack of trust in institutions and political representatives, nor that its application allows the fully achievement of the set of benefits that we present in Table 1. As Schudson [25] formulated, transparency is an essential element in democratic govern-ment but not a guarantee of good governance. We assume that access to public infor-mation is a democratic right that must be guaranteed [26]. Re-cipients do not form a uniform body, nor do governments behave according to the same logic, much less so at all levels of the administration, at every level, there may be instanc-es of power coexisting, sometimes led by very different forces, both within in each coun-try and in a comparative international scale. What is proposed is a model to build public information for a plural citizenship and as a source for an equally plural media in order to establish democratic and participatory political logics and processes. Thus, transparency is a good tool for immediate use and a seedling resource.

**Table 1.** Benefits of transparency.

| Area | Benefits |
|---|---|
| Democracy | Good governance<br>Accountability<br>Dialogue<br>Participation<br>Collaboration<br>Trust<br>Reputation |
| Knowledge | Synergies<br>Creation of models<br>Recognition of social plurality<br>Source for the media<br>Barrier against fake news |
| Development | Prevention of corruption<br>New opportunities<br>Improvements in management |

## 2. Objectives and Method

For active transparency to be more than an objective in itself, that is, for it to be useful for citizens, supporting and promoting democratic forms of participation, it must be approached and applied from a communicative perspective. The aim of this study is to determine the elements that define this perspective, namely principles, attributes and evidence of application.

Here we focus on the transparency of local public administrations, for which a method for analyzing transparency from the communicative perspective, the Infoparticipa method [27], was designed and applied in Spain. This method was applied to evaluate the transparency of the websites of local Spanish public administrations from 2013 to the present in annual waves, the results being published openly on the Infoparticipa Map [28]. This method was adapted to apply it also to the cases of Ecuador [29] and Colombia [30]. This involved studying the legislation regarding the organization of public administrations, transparency and the right of access to information in those countries, evaluating the transparency and quality of the information published on their websites and carrying out

a comparative analysis with other evaluation procedures of active transparency in local administrations of other countries based on diverse disciplinary approaches [31], using both published studies [32] and the sites where national and international organizations present their results and methods.

The communicative perspective employed in the Infoparticipa project separates the indicators from the method into two groups that were in turn divided into subgroups, addressing both the governance and management of the institutions and the way in which government actions are communicated and the way citizen participation is promoted. The groups are as follows: 1. Corporate transparency: (1.1) Who are the political representatives? (1.2) How do they manage collective resources? (1.3) How do they manage economic resources? 2. Information for participation: (2.1) What information do they provide regarding the municipal and the management of collective resources? (2.2) What tools do they offer for citizen participation? As can be seen, these groups of indicators address both governance and management issues as well as aspects related to the information produced by the institutions (group of indicators 2.1) and the procedures established to promote the democratic participation of citizens (group of indicators 2.2.).

Each area contains a group of indicators—which for Spain totaled 52—that specify the information to be published. Application criteria were established for each indicator and also published on the project website [28]. These application criteria determine the characteristics that the published information must have in order for the evaluation indicator to be positively validated. These criteria define what information must be found on the website as well as where it should have been published to be accessible, and other characteristics related to the understandability and timeliness of the information. Thus, any interested person (political representatives in the government or opposition, communication professionals from institutional cabinets or citizens in general) can access the evaluation indicators, the criteria with which they are applied, and the results obtained. These results are published on the Infoparticipa Map, a free and open-access tool that presents the results through interactive cartography.

The team of evaluators faces the evaluation as any other person would: the indicators are formulated in the form of a question that can only be answered positively to validate the indicator if all the required information is found on the web and complies with the required criteria. On the other hand, each indicator has the same percentage value over the total (100%). This is to facilitate an understanding of the results, so that anyone can in-terpret them. As can be appreciated, the methodology and procedure to communicate re-sults both seek to promote understanding, dissemination of results, and participation. For this reason, it is defined as a civic audit.

The indicators and criteria of the procedure were modified and updated based on the evaluation experience of the research team and the advice of a group of collaborating experts. Participating in this committee were representatives of the government of Catalonia and other provincial administrations, municipal organizations, the College of Journalists of Catalonia, universities, other professional or governmental organizations such as the Anti-Fraud Office, as well as university research group leaders.

In parallel, a research project was carried out on municipal communication offices, the results of which found that professionals and elected politicians have a low regard for the role of citizens and their possibilities of participation, as well as a persistent resistance to using public communication spaces to promote dialogue and deliberative processes [33,34]. On the other hand, a selection process has been made of good transparency practices on town hall websites in five strategic fields from the perspective of communication: the structure of the menu on the websites, essential to facilitate access to the websites' contents [35]; the links between different types of information, so that they complement each other for a better understanding and navigation [36]; accountability, essential for participation and democratic evaluation of government action [37]; the letters of service or documents which present the services provided along with the results of that provision of services and the future projects for those services [38]; and the use of news sections as

instruments at the service of transparency and accountability [39]. The conclusion of this last work is also essential for this project, since the existence of institutional communication practices that go beyond the mere publication of official documents has been verified. Sufficient examples have been found that demonstrate the interest of institutions and their political leaders in offering useful information to their citizens. These institutions and leaders are also interested in offering understandable information, presented in appropriate formats and published to favor dialogue and democratic deliberation.

As a result of all this, we established the theoretical and conceptual framework described in this article. This framework supports the Infoparticipa evaluation method and, at the same time, should guide the work of strategic institutional communication professionals and the perspective of the governments of public administrations to carry out effective and useful transparency practices for citizens.

## 3. Principles and Attributes of Transparency for Participation

### 3.1. Principles of Transparency

From the perspective of communication, transparency is a process in constant progress by which the institutions make all the information regarding their governance and management available to any person or organization such that any individual or legal subject can make use of it.

Its institutional application is essential because, in addition to being a democratic right, it results in different benefits that, from a communication perspective, we classify into three groups: democracy, knowledge and development.

In the first group, on the deepening of democracy, transparency is a pillar of good governance, facilitates accountability [40,41] and enables dialogue on equal terms to support certainty in deliberative processes [42] in which citizens can act individually or collectively through organizations or groups of stakeholders. As a consequence, it encourages participation [43] and collaboration, understanding the latter as a higher level in which citizens not only propose but also act together with leaders of the institutions. On the other hand, the provision of information and the experience of participation result in immaterial benefits such as trust in institutions [44–47] and improving their reputation [48].

Regarding the generation of knowledge, transparency fosters synergies with new agents and makes visible models of behavior, management and practices that can be replicated by other institutions. Moreover, the open provision of information leads to a recognition of social plurality, its characteristics and expectations [49]. To these we should add that transparency applied from the perspective of communication implies that institutions are reliable sources of information for journalism and the entire communication system, which currently includes the media and other multimedia and multiplatform possibilities in networks used by both professionals and citizens who interact with professional content or create new content. The correct functioning of this system depends on the fact that the main source of information, which comes from public institutions, is reliable, such that the wealth of information and opinions that it may generate becomes a barrier against fake news. Even authors like Schudson [25], who minimize the role of transparency in democracy, allude to the role of the media as amplifiers of institutional information.

Recent studies support changes in current journalistic routines in response to the emergence of fake news and the public's need for reliable news and media. Thus, Vu & Saldaña [50] considered that transparency and accountability should also govern journalistic practice, whereas Humprecht [51] studied the role of the so-called fact-checkers and points out the importance of carrying out their verifications in a transparent way, so that the public knows the process and therefore their results are credible. In both cases, it is stated that transparency must be based on the citation of sources and on the participation of the public in the production process. The public media, through transparent practices, must form these reliable sources on which the entire media system depends, and political participation must be based on complete and reliable information. Although this is not a definitive remedy against fake news, government transparency must act as a barrier and

collaborate as a positive element, which for its effectiveness must have a reliable political system as a whole.

Finally, transparency has a positive effect on the economy [52] because it acts as a preventive factor against corruption and because it improves the perception of other economic agents, thereby creating new opportunities. In addition, transparency has a direct consequence in terms of better management because it reviews the flow of information within the institutions and their practices.

However, for transparency to truly lead to these benefits, it must be based on the recognition of its importance by the institutions and the conviction of the people who work within them. In particular, those who have political and administrative responsibilities must be open to transparency. Therefore, transparency must be a shared value that forms part of the institutional culture so that it competes and pushes the entire collective to achieve it [53], thereby permeating institutional actions.

By value we mean an intrinsic idea—i.e., essential and permanent—that guides action. Thus, for transparency to be incorporated and put into practice by guiding and shaping institutional communication acts, it must be based on principles that lead to its creation as a conscious attitude. It is actually these principles that specify the meaning of transparency, appealing to conscience and coherence with manifest acts. Table 2 below sets out these principles.

**Table 2.** Principles of transparency.

| Principle | Content |
| --- | --- |
| Disclosure | Citizens have the right to know how public administrations are governed and managed. |
| Strengthening | Transparency strengthens the democratic system and the institutions that put it into practice. |
| Visibility | Information on the government and management of the institutions is a priority and must occupy prominent publication spaces. |
| Comprehensibility | Information that is not understandable is not transparent. |
| Dissemination | Information on institutional governance and management should not be discriminated against by prejudging which information will not be of interest or for whom it will not be of interest. |
| Humanism | Institutional transparency must make political and social pluralism visible. |

Disclosure. Institutional leaders have a legal obligation to report on how they govern and manage. However, above all they have a duty to do so as a consequence of their public function within the framework of the democratic system. The legal precept should not be the primary reason to be transparent since public administrations are at the service of the public and have their origin in the rule of law. They must therefore provide explanations that allow citizens to monitor their actions and base their decisions on political participation.

Strengthening. Transparency brings benefits for the entire democratic system, for society and for each transparent institution. In contrast, a lack of transparency and the partisan use of public communication damages the system and its institutions, eroding trust in both the short and long term. Transparency thus broadens political horizons, helping to promote participation as a goal of deepening the democratic system.

Visibility. Information related to the governance and management of the institution is the priority since it concerns how political representatives use their democratic mandate. It must therefore occupy the main spaces in the institutional media. In the case of institution websites, which are the main means of informing the public, transparency must be confined to an exclusive section that all too often is nothing more than a repository for documents that citizens avoid on the assumption that they will only find complex documents in it that may require considerable time and specific knowledge to read and understand. As is well known, when users are faced with too much information that is difficult to understand or that requires a high level of technical competence or a lot of management time, the relationship between transparency and trust is broken [54,55]. Thus, the website as a whole

must act as a tool for transparency, favoring knowledge, debate and participation and making it easier for citizens to access transparency information by navigating naturally, starting with the news sections. In this way, information on political decisions and the management of the institution reaches all citizens as part of the information flow. This does not preclude the possibility of publishing on other issues of interest or using other means such as social media to broaden the scope of information and reach other audiences. Indeed, all these communication means contribute to the dissemination of information, thereby offering possibilities for dialogue as a basis for participation.

Comprehensibility. We cannot speak of transparency if users do not understand the information because, even if the information has been published, there is no communication with the recipients. Consequently, institutional information, often technical and complex, should be published using journalistic language and resources and data visualization tools, audiovisual tools and all kinds of multimedia resources, provided they are useful.

Dissemination. Information on public management is of interest to all affected people, that is, to all citizens, though not all of them equally or for the same reasons or at the same time, as it is not possible to know precisely or in advance who the interested audiences will be. The publication formats or even the subjects themselves cannot be prepared by conditioning profiles of preferred users, as this results in some of the public being excluded from accessing or taking an interest in the information. This does not, however, mean that general publications cannot be complemented by specific ones aimed at particular groups, thus amplifying the dissemination of information or stimulating the participation of particularly interested groups.

Humanism. Today's societies are formed by a plural public made up of women and men of different origins, ages, social positions, interests, abilities, etc. [56]. This plurality should be manifested in the information if it is truly expected to stimulate interest and offer the information in a way that is appropriate to real social expectations. In this area, civic organizations play an important role in the democratic system and are instruments of collective organization and participation. Their interests and proposals must therefore be made visible. This includes making visible in the information the different political options and providing them spaces for expression. Elected political leaders have the right to use public communication mechanisms to publicize and explain their actions during their mandate. However, at the same time, citizens must know how elected politicians who do not form part of the government—who also have the right to make their proposals and interventions known so that they can be approved—exercise their opposition work. This means that they must be provided with spaces to explain themselves and that their interventions should be included in current spaces.

These principles must permeate the work of information production and the institutional culture so that transparency is understood and applied in such a way that the aim of informing to promote political participation is achieved.

### 3.2. Attributes of Transparent Information

In order for citizens, either individually or through formally or informally organized groups, to make productive use of the information, to achieve the benefits explained above and for public administrations to apply the principles of transparency described, the information must comply with attributes that are verifiable through evidence. Some of the attributes are cited ambiguously in certain national laws [57] but are presented separately from the articles in which the publication obligations are specified. As such, they are ineffective because they are seen only as declarations of intent.

Table 3 shows the attributes and principles of transparent information and the evidence of their application.

Veracity. This answers the question regarding what information is published, guaranteeing that the information is published in its entirety without omitting data or information in a self-serving way, something which would undermine the basis of transparency and the legitimacy of the institution. This implies making public the decisions made in all

the governing bodies and on the management in all areas, publishing the corresponding complete and approved documents with the appropriate documentary references for verification. These source texts, prepared by the corresponding bodies, certify the documentary basis of current information and its origin in democratic bodies and procedures. Given the complexity of many of these documents, they must be accompanied by information that makes it easier for the general public to understand (the "intelligibility" attribute).

**Table 3.** Attributes of transparent information.

| Question | Attribute | Principles That It Develops as a Priority | Requirement | Evidence |
| --- | --- | --- | --- | --- |
| What | Veracity | Disclosure, Strengthening and Dissemination | Complete information | Information on the decisions of all bodies and on the management of all areas. Complete and approved documents. |
| When | Timeliness | Visibility | Updated information | Current and continuously updated publications. Dated documents and publications appropriate to deadlines and actions. |
| Where | Accessibility | Visibility | Enabling website structure | Adequate content trees and understandable menus. Access to information in a maximum of 3 steps. |
| How | Usability | Visibility | Publication in usable, shareable formats | Publications in open, standardized, free and license-free formats. |
| | Intelligibility | Comprehensibility and Humanism | Information understandable to non-expert users | Journalistic treatment of information. Use of textual and audiovisual formats. Use of current news spaces. |
| | Universality | Comprehensibility and Humanism | Accessible, understandable information for all types of users | Publishing in greater levels of detail. Use of universal access standards. |
| Who | Pluralism | Disclosure, Strengthening and Humanism | Visibility of all policy options | Web publishing spaces of the different political options. Information on the proposals of the different options in the governing bodies. Diversity of sources. |
| | Plurality | Strengthening and Humanism | Visibility of social plurality | Visibility of the different groups and social organizations in the textual and audiovisual information. Diversity of sources. Humanization of information. |

For these documents to reach heads of communication, proper internal information management is necessary. By this we mean the ordering of internal information flows and the automation of statistical or documentary publication processes [49]. The ordering of information flows is essential because transparency is a practice that involves all professionals and all work areas of an institution, as well as everyone responsible for its governance and management. The information must reach heads of communication at the appropriate times so that the communication department receives the information, processes it and makes decisions regarding the most appropriate form of publication. For this, the commitment, preparation and obligation of everyone involved is vital [5].

On the other hand, the correct automation of statistical or documentary publication processes means that the primary documents and statistical information can be updated on the websites immediately by the areas that produce them. Trials have already been

conducted with the intervention of organizations such as the Consorci Administració Oberta de Catalunya [58], which provide support so that with a single action, administrative documents can be uploaded to the websites of control organizations while being published on the transparency portals of local administrations. Artificial intelligence should also be studied in order to determine how it might contribute to transparency, considering both its efficiency and ethical issues regarding algorithmic transparency itself [59,60].

Timeliness. This answers the question regarding when information is published. Timely information is that which is published when it is useful for the public and is updated whenever appropriate so that citizens can participate, having information that supports opinions and proposals appropriate to reality. A common malpractice is to publish or update the information once a year, forgetting that, if the aim is to keep the public informed about the issues affecting them and to promote their participation based on knowledge, it is necessary to inform at the appropriate time, namely when the information is current and contributes to reflection and debate. Documents and publications must therefore be dated, showing that they meet deadlines and legal actions and that they have been published in a timely manner.

Accessibility. This answers the question regarding where information is published. Transparent information must be easily accessible and any individual must be able to access it using the menus and tools on the website in a limited number of steps. The websites must facilitate navigation with a clear structure and using understandable headings. Otherwise, citizens end up giving up, with the repercussions this has for trust between the parties. However, institutional websites are not usually easy to navigate. The complexity of the menus and a poor definition of the content trees do not allow users to anticipate what they will find in each section or how to locate the information they are looking for in fewer steps. However, access is the basic premise for the appropriation of content and therefore the menu headings must be clear, grouping recognizable areas of information and ensuring that transparency is the structuring axis of the content and that the interaction of citizens with elected representatives and participation are present in all sections.

Usability. This answers the question regarding how information is published. For statistical and documentary information to be used by any interested person or organization, it must be presented in open, free and reusable formats that can be easily downloaded and used through standardized, commonly used operating systems and programs. It should therefore be possible to share the information and reuse it for purposes not anticipated even by those who prepared it. Therefore, the information must not be blocked so that it cannot be copied or reused. Indeed, to have an active public and enable participation, information needs to be easily reused and not protected by licenses or payments that limit its use.

Intelligibility. This also answers the question regarding how information is published but in terms of the intelligibility of the content. While governments have the right and duty to make their actions known, public media cannot be used as spaces for propaganda. Since most of the documents prepared by public administrations refer to legal, administrative and economic procedures, the source documents must be explained by applying a journalistic treatment and remembering that the best way of bringing the information closer to the public is in the form of news [61]. Sometimes an appropriate title or a title and an explanatory paragraph is sufficient. However, on other occasions a better treatment that includes maps, graphs or other infographics etc. is necessary. On the other hand, multimedia formats bring content closer to the public. The use of videos, infographics, podcasts, etc. for the presentation of data and the visualization and explanation of information makes it easier for the public to understand complex information such as economic and statistical information. At the same time, they are also powerful tools to demonstrate social plurality and political pluralism. Similarly, current news sections, which are the most visited spaces on institutional websites, must fundamentally show how institutes are governed. News sections are also flexible publication spaces that allow quick updates in

order to comply with the principle of timeliness and that they are published in journalistic format, facilitating access and understanding of their contents.

Universality. This also addresses the question regarding how information is published but considers the needs that people may have in accessing written or audiovisual information, considering any type of social, physical or cognitive characteristic [62]. So that any person is able to appropriate information regardless of different capacities and possibilities, public administrations must be exemplary. These users may be inexperienced people, the elderly, those with few economic resources or with physical or cognitive difficulties to whom accessibility, navigation, readability, appropriate formats of documents and forms, and usable multimedia elements must be provided [62]. For this, recognized procedures such as WCAG [63] or international easy-to-read criteria [64] must be used.

This attribute reinforces the need to offer information at different levels of detail so that individuals, from experts to others who only want to find out a specific aspect with a specific purpose, can access content. It is necessary to address both a public that expects to obtain essential information via a headline and a short text or other general access communication elements such as audiovisuals or infographics, and another specialized or especially motivated public who require complete and detailed information.

Pluralism. This answers the question regarding who the subjects of the information are, considering that the various political parties, in government or in opposition, have responsibilities and therefore must be subject to public scrutiny. Proposals of groups that do not form part of the government must be known and explained through compared and verifiable information. At the same time, they must have spaces for expression on institutional websites. On the other hand, the communicative approach involves also addressing the fundamental practices of journalism, highlighting among these the diversity of sources, which should not be exclusively governmental [65].

Plurality. This also answers the question regarding who the subjects of the information are but considering the obligation to make social plurality visible through the representation of different groups and organizations. For this, it is vital that they also appear as sources of information and as subjects of the texts and images. This implies the need to humanize the information [49], which means that for the receiving community to be able to appropriate the information and use it, it has to present and question it as a human being, taking into account the diversity of people who make up the social body and who read the written or audiovisual texts, interpreting them from their reality and comparing them with their life experience and interests.

## 4. Discussion

While finishing this article, we witnessed the storming of the Capitol of the United States of America (6 January 2021). The institutional desecration this event entailed demonstrates, once again, that democratic institutions around the world are not safe from retrogression as a result of rising populist policies, which challenge the democratic system by spreading fake news, attacking the media system as a whole and launching messages that erode the very institutions from which power is exercised [66]. It is vital that democratic institutions strengthen their legitimacy and reputation as guarantors of coexistence and freedoms. Therefore, it is vital that citizens have trust in transparent, inclusive public communication. We approach this problem, which has regained relevance and importance, by showing that the communicative perspective of transparency is adequate to address these needs.

Today, transparency is a powerful idea that must be exploited and is decisive in terms of the public consideration of the institutions and the people who govern them, with transparency and opacity being used as discursive instruments of political legitimation or delegitimization and transparency being presented as an essential quality of good governance [67]. Despite this, it is often a rhetorical discourse, an artificial instrument in the political dispute, empty of content, yet paradoxically effective since it is always possible to ask for greater transparency or accuse institutes of a lack of transparency. Although

studies can set margins for sufficient compliance for active transparency, the degree of completeness will always be controversial in political confrontation.

Resources and efforts are often used to publish information only in order to comply with mandatory regulations, to avoid accusations of opacity and, consequently, suspicions of corruption, or to present themselves as champions of transparency, as if its formal practice were sufficient. To prevent transparency becoming a political rhetorical device, for it to be effective in anticipating and supporting participation, efficient communication must be carried out considering the principles of transparency as a value and applying its attributes, recognizable in evidence, in practice. For this to be possible, it is necessary for political and administrative leaders to adopt a proactive attitude. Political leadership is essential to promote a transparent administration and to support the role of journalists and communicators as professionals trained for this task.

From the perspective of communication, transparency addresses the challenges involved in using it as a way to achieve the political participation of citizens. However, even before the appearance of the Internet, public media organizations were often used for the exclusive benefit of the government of the day [68] and despite the enormous possibilities of digital and online media platforms in facilitating interaction with the public, political participation [69] and even building a new knowledge paradigm [56], the approaches on which institutional communication is designed continue to be anchored in traditional practices.

## 5. Conclusions

In this article we have addressed active transparency, establishing the framework from which to approach research based on the understanding of transparency as a communicative process, overcoming practices that prevent progress towards new forms of deliberative democracy. However, together with active transparency, administrations must also respond to requests for information, something which also forms part of proper communication behavior.

On the other hand, the implementation of transparency as an instrument for participation in each country takes particular forms, as a result of diverse political, economic or technological trajectories. Thus, the model we propose can be used universally but not synchronously. Its application must allow public administrations to better understand the importance of transparency, develop adequate policies for its implementation and advance in the generation of a transparent institutional culture.

Finally, it should be noted that we have focused on local administrations and as such, it will be necessary to study how to apply this framework to supralocal administrations, which have different competences, needs and traditions.

**Author Contributions:** Conceptualization, Pedro Molina Rodríguez-Navas; Data curation, Pedro Molina Rodríguez-Navas; Formal analysis, Pedro Molina Rodríguez-Navas; Investigation, Pedro Molina Rodríguez-Navas, Narcisa Medranda Morales and Johamna Muñoz Lalinde; Methodology, Pedro Molina Rodríguez-Navas, Narcisa Medranda Morales and Johamna Muñoz Lalinde; Resources, Pedro Molina Rodríguez-Navas; Supervision, Pedro Molina Rodríguez-Navas; Validation, Pedro Molina Rodríguez-Navas; Writing—original draft, Pedro Molina Rodríguez-Navas, Narcisa Medranda Morales and Johamna Muñoz Lalinde; Writing—review & editing, Pedro Molina Rodríguez-Navas. All authors have read and agreed to the published version of the manuscript.

**Funding:** This research received no external funding.

**Institutional Review Board Statement:** Not applicable.

**Informed Consent Statement:** Not applicable.

**Conflicts of Interest:** The authors declare no conflict of interest.

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
