# Peer review of "Transparency for Participation through the Communication Approach"

_ijgi, doi:10.3390/ijgi10090586_

Round 1
Reviewer 1 Report
A very interesting article providing a comprehensive model of principles. However, I have not found a link between how application of this model may counteract fake news. The example provided in the first paragraph of Section 4 shows that no matter how transparent government information may be, citizens may still choose to believe a parallel truth provided by other sources if it suits their perception. There are examples of more transparency not necessarily leading to more trust in government and even having the opposite effect: leading to more mistrust as government organizations can be viewed to fail in the perception of the public. In addition, your article describes that you used three Spanish language administrations to come up with your conceptual model. Do you think your conceptual model is universal enough to be used in other countries?

Author Response
1. I have not found a link between how application of this model may counteract fake news.
We have expanded the explanation on this subject in lines 253-265, expanding our first explanation in lines 243-252:
"Recent studies support changes in current journalistic routines in response to the emergence of fake news and the public's need for reliable news and media. Thus, Vu & Saldaña (2021) considered that transparency and accountability should also govern journalistic practice, whereas Humprecht (2020) studied the role of the so-called fact-checkers and points out the importance of carrying out their verifications in a transparent way so that the public knows the process and therefore their results are credible. In both cases, it is stated that transparency must be based on the citation of sources and on the participation of the public in the production process. The public media, through transparent practices, must form these reliable sources on which the entire media system depends, and political participation must be based on complete and reliable information. Although this is not a definitive remedy against fake news, government transparency must act as a barrier and collaborate as a positive element, which for its effectiveness must have a reliable political system as a whole."
2. Do you think your conceptual model is universal enough to be used in other countries?
Our answer to this question is in lines 547-552:
"On the other hand, the implementation of transparency as an instrument for participation in each country takes particular forms, as a result of diverse political, economic or technological trajectories. Thus, the model we propose can be used universally but not synchronously. Its application must allow public administrations to better understand the importance of transparency, develop adequate policies for its implementation and advance in the generation of a transparent institutional culture."
Reviewer 2 Report
The article addresses a topic of recognized interest and that assumes growing scientific relevance. It presents an interesting contribution to studies on transparency, an area in development, so it is very relevant and current.
Its greatest contribution lies in the communicational approach chosen by the authors who carry out a characterization of transparency that encompasses several dimensions, which establish a fruitful dialogue between the normative dimension and the practical application of the concept. Also relevant is the fact that this is an investigation applied to the political communication of the local public administration, in a perspective of deepening the participation that takes into account that it is not enough to provide information, but that it is essential that citizens are able to act reflexively based on in that information.
The bibliography is adequate and updated. Although the article does not focus on the epistemological discussion of the concept of transparency, there is a gap due to the scarcity of references (we recorded essentially the reference to Michael Schudson, 2020) regarding the problematization of the ambivalences of transparency in democracy, in the line of studies such as Amitai Etzioni's (2010), Is Transparency the Best Disinfectant?; by Mark Fenster (2015), Transparency in search of a theory; or approaches of a critical nature such as those gathered in the work edited by Emmanuel Alloa and Dieter Thomä (2018), Transparency, Society and Subjectivity. This gap must be addressed.
Author Response
1. Although the article does not focus on the epistemological discussion of the concept of transparency, there is a gap due to the scarcity of references.
We have added the suggestions made by the reviewer in lines 103-124:
"On the other hand, Etzioni (2010) stated that – in the best of cases – the supposed benefits of transparency are limited, and the usefulness of the information does not compensate for its high cost. Fenster (2015) also argued that theory about transparency has not been sufficiently developed and that it starts from assumptions that must still be clarified to get rid of idealized concepts not tested in practice. Alloa and Thöma (2018) mentioned that we can talk of the field of Critical Transparency Studies, which has emerged due to the abusive and ambiguous use of the term ‘transparency’ and because it has also been noted that public administrations often use it as a marketing strategy rather than as an effective strategy.
In this work, it is not assumed that transparency by itself solves all kinds of problems related to the lack of trust in institutions and political representatives, nor that its application allows the fully achievement of the set of benefits that we present in Table 1. As Schudson (2020) formulated, transparency is an essential element in democratic government but not a guarantee of good governance. We assume that access to public information is a democratic right that must be guaranteed (Bertot, Jaeger & Grimes, 2010). Recipients do not form a uniform body, nor do governments behave according to the same logic, much less so at all levels of the administration, at every level, there may be instances of power coexisting, sometimes led by very different forces, both within in each country and in a comparative international scale. What is proposed is a model to build public information for a plural citizenship and as a source for an equally plural media in order to establish democratic and participatory political logics and processes. Thus, transparency is a good tool for immediate use and a seedling resource."
Reviewer 3 Report
Muchas gracias por el texto, que ahonda en un tema de vital importancia para las administraciones públicas.
Entiendo que el manuscrito requiere un trabajo adicional, sobre todo de cara a brindar una explicación mucho más detallada de su diseño metodológico: el método Infoparticipa no tiene mucho más que un link que se presenta para recabar mayor información, y entiendo que debería quedar claro en el texto sin necesidad de recurrir a sitios externos. Con ello, considero, se entendería mejor la presentación de los resultados.
Además, me permito sugerir otras puntualizaciones:
. En el abstract, líneas 17 y 18, entiendo que un "could" sería más conveniente que el "can"actual; líneas 18 a 21, la metodología debería estar más claramente explicitada.
. En la introducción, se necesita una mejor explicación que contextualice adecuadamente desde el principio que el fenómeno de la transparencia se aplica al sector público y a su comunicación pública.
. Línea 146. Entiendo que el término bimembre "institutional communication" no es de uso habitual en el habla inglesa, con lo cual sugiero reemplazarlo por public relations y/o strategic communication.
. Sería deseable estructurar una sección dedicada a consideraciones finales o conclusiones, en la que sería deseable ampliar las recomendaciones y limitaciones del estudio.
Finalmente, alguna caracterización en la sección de literature review sobre conceptos muy cercanos al de transparencia (citizen satisfaction, citizen engagement, public legitimacy, trust, reputation) podría ser de utilidad para enriquecer el manuscrito; textos como el de Canel y Luoma-aho (Public Sector Communication, Wiley, 2019) aportan miradas valiosas.
Author Response
1. El método Infoparticipa no tiene mucho más que un link que se presenta para recabar mayor información, y entiendo que debería quedar claro en el texto sin necesidad de recurrir a sitios externos.
We have expanded considerably the description of the methodology with some changes in the writing which clarify the section (lines 146-147) and new paragraphs in lines 155-158, 163-180 and 190-206.
The communicative perspective employed in the Infoparticipa project separates the indicators from the method into two groups that were in turn divided into subgroups, addressing both the governance and management of the institutions and the way in which government actions are communicated and the way citizen participation is promoted. The groups are as follows: 1. Corporate transparency: (1.1) Who are the political representatives? (1.2) How do they manage collective resources? (1.3) How do they manage economic resources? 2. Information for participation: (2.1) What information do they provide regarding the municipal and the management of collective resources? (2.2) What tools do they offer for citizen participation? As can be seen, these groups of indicators address both governance and management issues as well as aspects related to the information produced by the institutions (group of indicators 2.1) and the procedures established to promote the democratic participation of citizens (group of indicators 2.2.).
Each area contains a group of indicators —which for Spain totaled 52— that specify the information to be published. Application criteria were established for each indicator and also published on the project website (see LabComPublica, n.d.). These application criteria determine the characteristics that the published information must have in order for the evaluation indicator to be positively validated. These criteria define what information must be found on the website as well as where it should have been published to be accessible, and other characteristics related to the understandability and timeliness of the information. Thus, any interested person (political representatives in the government or opposition, communication professionals from institutional cabinets or citizens in general) can access the evaluation indicators, the criteria with which they are applied, and the results obtained. These results are published on the Infoparticipa Map, a free and open-access tool that presents the results through interactive cartography.
The team of evaluators faces the evaluation as any other person would: the indicators are formulated in the form of a question that can only be answered positively to validate the indicator if all the required information is found on the web and complies with the required criteria. On the other hand, each indicator has the same percentage value over the total (100%). This is to facilitate an understanding of the results, so that anyone can interpret them. As can be appreciated, the methodology and procedure to communicate results both seek to promote understanding, dissemination of results, and participation. For this reason, it is defined as a civic audit.
The indicators and criteria of the procedure were modified and updated based on the evaluation experience of the research team and the advice of a group of collaborating experts. Participating in this committee were representatives of the government of Catalonia and other provincial administrations, municipal organizations, the College of Journalists of Catalonia, universities, other professional or governmental organizations such as the Anti-Fraud Office, as well as university research group leaders.
In parallel, a research project was carried out on municipal communication offices, the results of which found that professionals and elected politicians have a low regard for the role of citizens and their possibilities of participation, as well as a persistent resistance to using public communication spaces to promote dialogue and deliberative processes (Rodríguez-Breijo et al. 2021: Simelio et al. 2019). On the other hand, a selection process has been made of good transparency practices on town hall websites in five strategic fields from the perspective of communication: the structure of the menu on the websites, essential to facilitate access to the websites’ contents (Molina & Corcoy, 2019); the links between different types of information, so that they complement each other for a better understanding and navigation (Molina, 2019a); accountability, essential for participation and democratic evaluation of government action (Molina, 2019b); the letters of service or documents which present the services provided along with the results of that provision of services and the future projects for those services (Molina, 2019c); and the use of news sections as instruments at the service of transparency and accountability (Molina, 2019d). The conclusion of this last work is also essential for this project, since the existence of institutional communication practices that go beyond the mere publication of official documents has been verified. Sufficient examples have been found that demonstrate the interest of institutions and their political leaders in offering useful information to their citizens. These institutions and leaders are also interested in offering understandable information, presented in appropriate formats and published to favor dialogue and democratic deliberation.
2. En el abstract, líneas 17 y 18, entiendo que un "could" sería más conveniente que el "can"actual.
We have made the suggested change in lines 17 and 18: "...although most of the concepts could be applied at other levels of public administration".
3. En el abstract, la metodología debería estar más claramente explicitada".
We have added an explicit reference to the Infoparticipa methodology (line 20): "To establish this framework, the legislation and application of transparency in three countries (Spain, Ecuador, and Colombia) was studied using the Infoparticipa method designed with a communication approach in mind"
To be able to do this, we have erased a part of the abstract. It is not possible to explain more about the methodology without exceeding the abstract´s 200-word limit. We can only exceed this word limit if we receive authorization from the editor.
4. Línea 146. Entiendo que el término bimembre "institutional communication" no es de uso habitual en el habla inglesa, con lo cual sugiero reemplazarlo por public relations y/o strategic communication.
We have made this change, now in line 211: "This framework supports the Infoparticipa evaluation method and, at the same time, should guide the work of strategic institutional communication professionals and the perspective of the governments of public administrations to carry out effective and useful transparency practices for citizens".
5. Sería deseable estructurar una sección dedicada a consideraciones finales o conclusiones, en la que sería deseable ampliar las recomendaciones y limitaciones del estudio.
We have created a new section 5 titled “Conclusions” (line 534) in which we have expanded the limitations of the study (lines 541-555): " In this article we have addressed active transparency, establishing the framework from which to approach research based on the understanding of transparency as a communicative process, overcoming practices that prevent progress towards new forms of deliberative democracy. However, together with active transparency, administrations must also respond to requests for information, something which also forms part of proper communication behavior.
On the other hand, the implementation of transparency as an instrument for participation in each country takes particular forms, as a result of diverse political, economic or technological trajectories. Thus, the model we propose can be used universally but not synchronously. Its application must allow public administrations to better understand the importance of transparency, develop adequate policies for its implementation and advance in the generation of a transparent institutional culture.
Finally, it should be noted that we have focused on local administrations and as such, it will be necessary to study how to apply this framework to supralocal administrations, which have different competences, needs and traditions."
6. Alguna caracterización en la sección de literature review sobre conceptos muy cercanos al de transparencia (citizen satisfaction, citizen engagement, public legitimacy, trust, reputation) podría ser de utilidad para enriquecer el manuscrito; textos como el de Canel y Luoma-aho (Public Sector Communication, Wiley, 2019) aportan miradas valiosas.
We have introduced the suggested text briefly in lines 48-53: "María José Canel and Vilma Luoma-aho (2019) presented how changes in demands, expectations, forms of communication, roles and even the very nature of citizenship affect the public sector. Thus, new challenges arise such as responding to demands or achieving citizen engagement with institutions, to achieve an increase in intangibles aspects such as reputational or a greater perception of legitimacy".
Other references to the suggested concepts:
- Legitimacy: lines 387 and 506
- Trust: table 1, and lines 113, 237, 294, 306, 429, 507.
- Reputation: table 1, and lines 238 and 506.